# Investigation of the Acute Effects of Two Different Preoxygenation Methods on Neurodegenerative Biomarkers in Laparoscopic Cholecystectomy Surgery

**DOI:** 10.3390/medicina61020167

**Published:** 2025-01-21

**Authors:** Veli Fahri Pehlivan, Basak Pehlivan, Hakim Celik, Erdogan Duran, Abdullah Taskın, Seyhan Taskın, Faik Tatlı

**Affiliations:** 1Department of Anesthesia and Reanimation, Faculty of Medicine, Harran University, Osmanbey Campus, PC 63300 Sanliurfa, Turkey; bpehlivan@harran.edu.tr (B.P.); drerdoganduran6331@gmail.com (E.D.); 2Department of Physiology, Faculty of Medicine, Harran University, Osmanbey Campus, PC 63300 Sanliurfa, Turkey; hakimcell@gmail.com (H.C.); seyhan_taskin@yahoo.com (S.T.); 3Department of Nutrition and Dietetics, Health Science Faculty, Harran University, Osmanbey Campus, PC 63300 Sanliurfa, Turkey; abdullahtaskin52@harran.edu.tr; 4Department of General Surgery, Faculty of Medicine, Harran University, Osmanbey Campus, PC 63300 Sanliurfa, Turkey; faiktatli-@hotmail.com

**Keywords:** preoxygenation, anesthesia, neurodegenerative biomarkers (pTau, S100B, NSE, GFAP), neurotoxicity, hyperoxia

## Abstract

*Background and Objectives*: Oxygen is essential for all living organisms and plays a critical role in anesthesia and intensive care practices. However, the notion that unlimited oxygen therapy is harmless is a misconception. Our study investigates the acute effects of different preoxygenation methods on hemodynamic parameters and neurodegenerative biomarkers in patients undergoing laparoscopic cholecystectomy surgery. *Materials and Methods*: This prospective, randomized, controlled study included 52 patients undergoing elective laparoscopic cholecystectomy under general anesthesia. Patients were divided into two groups: Group I received standard preoxygenation (100% FiO_2_ for 3 min), while Group II underwent rapid preoxygenation (eight deep breaths over 30 s to 1 min). Hemodynamic parameters (SAP, DAP, MAP, and SpO_2_) and neurodegenerative biomarkers (pTau, S100B, NSE, NfL, GFAP) were measured after preoxygenation, after intubation, and at the end of surgery. *Results*: Group I exhibited a significant increase in levels of pTau, S100B, NSE, and GFAP, indicating higher neuronal and glial cell stress compared to Group II (*p* < 0.001). No significant increase in NfL levels was observed in either group. Hemodynamic parameters (HR, SAP, DAP, MAP) were significantly higher during and after preoxygenation in Group I, suggesting an increased stress response. Group II showed lower levels of acute neurotoxicity and oxidative stress. *Conclusions*: Our findings indicate that preoxygenation with 100% FiO_2_ induces stress in neuronal cells, axons, and glial cells, leading to an increase in neurodegenerative biomarkers. Optimizing preoxygenation strategies is crucial to reduce oxidative stress and improve neurological outcomes for surgical patients.

## 1. Introductions

Oxygen is essential for the survival of all living organisms and plays an indispensable role in anesthesia and intensive care practices. However, the assumption that unrestricted oxygen use is harmless is misguided. Evidence from various clinical scenarios suggests that unrestricted and/or high-concentration oxygen administration can have harmful effects. Therefore, therapeutic goals must be re-evaluated to ensure both the safety and efficacy of oxygen therapy [1,2]. Oxygen therapy should be meticulously tailored to the right patient, at the right time, and at the correct dosage to prevent the adverse outcomes of both hypoxia and hyperoxia.

Hyperoxia is frequently observed in patients undergoing surgery under anesthesia or in intensive care. Its acute toxicity arises from the increased production of reactive oxygen species (ROS) due to oxidative stress, leading to macromolecular damage and disrupted cellular signaling pathways. ROS generated by hyperoxia oxidize lipids, DNA, and proteins, causing widespread oxidative damage that can result in severe and sometimes permanent pathologies [3,4,5]. Complications of preoxygenation include delayed detection of esophageal intubation, absorption atelectasis, ROS production, and undesirable hemodynamic effects [6,7]. While avoiding both hypoxemia and hyperoxia is critical, the optimal oxygen dose for use in intensive care and anesthesia remains uncertain. Moreover, it is unclear whether conservative oxygen therapy reduces overall mortality, as its effects may vary among patient populations [8]. Nevertheless, the benefits of preoxygenation in preventing hypoxemia, especially in patients with anticipated difficult airways, outweigh its potential risks and justify its use.

Neuronal proteins, which serve as biomarkers of central nervous system damage, are critical tools in understanding the impact of anesthesia and surgical stress on neuronal integrity. These biomarkers are released into the plasma in response to acute injuries such as trauma, concussion, and ischemia [9], providing a real-time snapshot of neurological stress. The temporal profiles of these biomarkers vary, with each displaying distinct surges corresponding to the acute, subacute, and chronic stages of injury progression (Figure 1) [10,11]. For example, the neurofilament light chain (NfL), a sensitive indicator of axonal injury, rises predominantly during the later stages [12,13], while neuron-specific enolase (NSE), an enzyme indicative of neuronal injury, is elevated in the acute phase [14].

High oxygen levels during anesthesia have been linked to poor neurological outcomes, including increased NSE levels [15]. Similarly, tau proteins, integral to the stability of neuronal microtubules, undergo hyperphosphorylation under oxidative stress conditions such as hyperoxia, leading to elevated pTau levels that signify potential neurodegenerative processes [16,17]. S100B, a calcium-binding protein expressed by astrocytes, also rises after oxygen exposure and has been associated with poor cognitive outcomes in surgical patients [18,19]. Astrocytes, the primary constituents of the blood–brain barrier, may increase GFAP production in response to oxidative stress, reflecting their role in maintaining homeostasis and responding to injury [20].

These biomarkers are increasingly recognized as crucial indicators of anesthesia-induced neurotoxicity and surgical stress. The observed elevations in neurodegenerative markers following surgery suggest that general anesthesia and associated oxidative stress may contribute to short-term neuronal damage, potentially setting the stage for long-term neurological sequelae if not carefully managed [16,21]. This underscores the importance of integrating biomarker monitoring into anesthetic practices to guide neuroprotective strategies and minimize risks.

Preoxygenation is a maneuver designed to increase oxygen reserves and delay arterial hemoglobin desaturation during apnea. While not mandatory, it is recommended for all patients before anesthesia [6,22,23]. Ventilation and intubation challenges are often unpredictable, making preoxygenation an essential precautionary measure [6,22,24]. Although anesthetic agents are often regarded as the primary contributors to cognitive decline due to their direct effects on the brain, oxygen has also emerged as a significant factor in this process [25]. High FiO_2_ administered preoperatively or intraoperatively may exacerbate oxidative stress, inflammation, and toxicity [21]. Furthermore, prolonged oxygen exposure is known to cause central nervous system toxicity [26,27].

Standard preoxygenation with 100% FiO_2_ typically involves tidal volume breathing (TVB) for at least three minutes, while rapid preoxygenation—eight deep breaths within 30 s to one minute—can achieve similar efficiency [28,29,30]. Positive airway pressure techniques (e.g., NIV, CPAP, PEEP, BIPAP) or high-flow nasal oxygen (HFNO) can also be employed. The choice of technique should be individualized based on the patient and clinical factors as well as the available equipment [23].

This study aims to investigate the acute effects of different preoxygenation methods on hemodynamic parameters and neurodegenerative biomarkers in patients undergoing laparoscopic cholecystectomy. Our hypothesis is that rapid preoxygenation, delivering less oxygen than standard preoxygenation, will reduce oxidative stress and result in lower neurodegenerative biomarker levels. By implementing preoxygenation for a shorter duration and with a lower oxygen dose, exposure to high oxygen concentrations is minimized, potentially reducing oxidative stress-related side effects. This hypothesis has the potential to contribute to the development of safer and more effective oxygen therapy strategies in anesthesia and intensive care practices.

## 2. Materials and Methods

### 2.1. Study Setting and Population

This prospective randomized controlled study was conducted on 52 patients with ASA (American Society of Anesthesiologists) physical status I–II who underwent elective laparoscopic cholecystectomy under general anesthesia. The sample size was calculated as 15 with an effect size of 0.722, with a margin of error of 0.05 and a power of 0.95, using correlation analysis, taking Amalia L et al.’s study as a reference [31]. This study was conducted in accordance with the Strengthening the Reporting of Observational Studies in Epidemiology (STROBE) guidelines. A total of 70 patients were invited to be included in the study (Group 1: 35, Group 2: 35). The study was conducted between June 2023 and February 2024, with patients recruited from the Department of General Surgery at Harran University (Figure 2).

All patients underwent laparoscopic cholecystectomy for chronic, symptomatic cholelithiasis confirmed by ultrasonography without signs of acute cholecystitis. Written and signed informed consent was obtained from all participants prior to inclusion in the study. The study protocol adhered to the principles of the Declaration of Helsinki and was approved by the Harran University Clinical Research Ethics Committee (Approval Date: 7 February 2022, Session No. 03, HRÜ: 22/03/11).

### 2.2. Randomization and Blinding

Randomization was performed using a computer-based random number generator to assign patients into two groups: Group I (standard preoxygenation with 100% FiO_2_ for 3 min) and Group II (rapid preoxygenation with eight deep breaths over 30 s to 1 min). This randomization procedure ensured unbiased participant allocation, minimized selection bias, and balanced potential confounding factors across the groups.

### 2.3. Preoxygenation Technique

Preoxygenation was performed in accordance with anesthesia and difficult airway guidelines [23]. Patients were divided into two groups. All participants underwent a standard preoxygenation technique prior to anesthesia. Prior to anesthesia induction, Group I underwent standard preoxygenation with 100% O_2_ for 3 min using the classic method (n = 30), while Group II performed rapid preoxygenation consisting of eight deep breaths within 30 s to one minute (n = 22).

### 2.4. Anesthesia Technique

All patients were prepared for anesthesia by establishing intravenous access and monitoring with an electrocardiogram, noninvasive blood pressure, pulse oximetry, and ET CO_2_. To ensure patient safety, oxygenation levels were continuously monitored using a pulse oximeter [32]. The following premedications were administered to all patients: 3 mg of midazolam and 50 mg of ranitidine hydrochloride. Induction was initiated with propofol (2 mg/kg), fentanyl citrate (2 μg/kg), and rocuronium bromide (0.6 mg/kg). Balanced anesthesia was maintained with 1–2% sevoflurane. After tracheal intubation, the lungs were ventilated with a 50/50 oxygen/air mixture. Ventilation was provided with a tidal volume of 6–8 mL/kg and a 50/50% oxygen and medical air mixture. The ventilation rate was adjusted to keep PaCO_2_ values between 35 and 40 mm Hg. A neuromuscular blockade was antagonized at the end of surgery with 0.05 mg/kg neostigmine methylsulfate and 0.02 mg/kg atropine sulfate. Postoperatively, all included patients received a 1 g paracetamol infusion every 8 h and 400 mg/4 mL IV ibuprofen every 12 h. No opioids were used postoperatively.

Venous blood samples (5 mL) were taken from the patients immediately after preoxygenation, before intubation (T1), at the 5th minute after endotracheal intubation (T2), and at the end of the surgical procedure (T3). During blood sampling, hemodynamic parameters, including heart rate (HR), systolic arterial pressure (SAP), diastolic arterial pressure (DAP), mean arterial pressure (MAP), and oxygen saturation (SpO_2_), were recorded. Whole blood samples were allowed to clot at room temperature for 1 h, followed by centrifugation at 1000× *g* for 20 min at 4 °C. After centrifugation, serum samples were separated and stored at −80 °C until analysis.

### 2.5. Neurodegenerative Biomarker Analysis

Serum S100B (cat. no. E-EL-H1297), NSE (cat. no. E-EL-H1047), NfL (cat. no. E-EL-H0741), pTau (cat. no. E-EL-H5314), and GFAP (cat. no. E-EL-H6093) levels were determined using commercial ELISA kits following the manufacturer’s instructions (Elabscience Biotechnology Inc., Wuhan, Hubei Province, China). All ELISA kits used in this research work with the Sandwich-ELISA principle. The micro ELISA plate provided in those kits is pre-coated with an antibody specific to human S100B (NSE, NfL, pTau and GFAP). Serum samples (or Standards) were added to the micro ELISA plate wells and combined with the specific antibody. Then, a biotinylated detection antibody specific for Human S100B (NSE, NfL, pTau and GFAP) and AvidinHorseradish Peroxidase (HRP) conjugate were added successively to each micro plate well and incubated. Artifacts were washed away. The substrate solution was added to each well. Only those wells that contain human S100B (NSE, NfL, pTau and GFAP), biotinylated detection antibody, and Avidin-HRP conjugate will appear blue in color. The enzyme-substrate reaction was terminated by the addition of stop solution and the color turned yellow. The optical density was measured with a microplate reader (Varioskan LUX; Thermo Fisher Scientific, Waltham, MA, USA) at a wavelength of 450 nm. Serum S100B (NSE, NfL, pTau and GFAP) concentration was calculated by comparing the OD of the samples with the standard curve. S100B, NfL, pTau, and GFAP levels were expressed as pg/mL, and NSE level was expressed as ng/mL.

### 2.6. Outcomes

#### 2.6.1. Primary Outcome

The primary outcome of this study was to evaluate the acute effects of two different preoxygenation methods on neurodegenerative biomarkers (pTau, S100B, NSE, NfL, GFAP) in patients undergoing laparoscopic cholecystectomy surgery.

#### 2.6.2. Secondary Outcome

Secondary outcomes included changes in hemodynamic parameters (SAP, DAP, MAP, SpO_2_) during preoxygenation, after intubation, and at the end of the surgery.

### 2.7. Statistical Analysis

Statistical analyses were performed using the SPSS 25.0 software package (IBM SPSS Inc., Chicago, IL, USA). The Shapiro–Wilk test was used to assess the normality of the data distribution. Numerical variables with a normal distribution were expressed as mean ± standard deviation, while numerical variables that did not follow a normal distribution were expressed as a median (interquartile range). Categorical variables were presented as a number (n) and percentage (%). The comparison of numerical variables between two groups was conducted using the independent samples *t*-test for normally distributed data and the Mann–Whitney U test for non-normally distributed data. For multiple dependent comparisons, Repeated Measures ANOVA and the Friedman test were used; Bonferroni correction was applied for post hoc pairwise comparisons. The Pearson Chi-square test was used to compare categorical variables. A confidence interval (CI) of 95% was accepted for all analyses, and a *p*-value < 0.05 was considered statistically significant.

## 3. Results

### 3.1. Demographic and Clinical Characteristics of the Patients

During the study period, 70 patients were invited to participate; however, 10 patients declined to participate, and 8 patients were excluded. Patients who experienced hemodynamic or ventilation problems during surgery (four patients), those who experienced shock or bleeding during the operation (two patients), and those with a history of alcohol consumption, regardless of the amount or frequency, either in the past or present (two patients), were excluded from the study. As a result, data were collected from 52 patients (Figure 2). In our study, the acute effects of two different preoxygenation methods (standard preoxygenation and rapid preoxygenation) on hemodynamic parameters and neurodegenerative biomarkers were examined.

There was no significant difference between the two groups in terms of patients’ demographic data, comorbidities, and baseline laboratory values. (Table 1 and Table 2).

### 3.2. Changes in Hemodynamic Parameters (SAP, DAP, MAP, SpO_2_)

Hemodynamic parameters were compared in patients who underwent standard preoxygenation with 100% O_2_ for 3 min before anesthesia induction (Group I) and rapid preoxygenation consisting of eight deep breaths within 30 s to one minute (Group II) at three time points: after preoxygenation (T1), post-endotracheal intubation (T2), and at the end of the surgical procedure (T3) (Table 3). In Group I, HR, SAP, DAP, and MAP levels measured at the three different time points were statistically significant, but SpO_2_ levels were not. HR levels at T1 and T2 were higher and statistically significant compared to T3 (*p* = 0.009 and *p* = 0.001, respectively). SAP levels measured at T1 were found to be higher and statistically significant compared to those at T2 and T3 (*p* = 0.011 and *p* = 0.003, respectively). While DAP levels did not show a significant change at T1 and T2, the DAP level at T1 was higher and significant compared to T3 (*p* = 0.020). Similarly, the MAP level at T1 was higher and significant compared to the MAP level at T3 (*p* = 0.019).

In Group II, significant changes were found between HR and SAP levels measured at the three different time points, while no statistical difference was observed in SpO_2_, DAP, and MAP levels. In Group II, the HR level measured at T2 was higher and significant compared to T3 (*p* = 0.023). The SAP level measured at T1 was the highest among the three time points, and this was statistically significant compared to the T2 level (*p* = 0.009). These changes were evaluated as responses to increased stress (Table 3).

### 3.3. Evaluation of Acute Effects of Preoxygenation Methods on Neurodegenerative Biomarkers (pTau, S100B, NSE, NfL, GFAP)

It was found that the levels of pTau, S100B, NSE, and NfL measured at three different time points in the serum samples of patients included in Group I and Group II did not show any significant changes (Table 4) (*p* > 0.05). However, in both groups, GFAP levels were found to decrease over time, which was statistically significant (*p* = 0.011). The GFAP level at any time point in Group I was twice as high as the GFAP level measured at any time point in Group II. In Group I, GFAP levels measured at the three different time points were statistically significant. The GFAP level at T1 was higher and statistically significant compared to that at T2 and T3 (*p* = 0.047 and *p* = 0.009, respectively). No significant change was observed between T2 and T3. In Group II, GFAP levels measured at the three different time points were also statistically significant. The GFAP level at T1 was higher and statistically significant compared to that at T2 and T3 (*p* = 0.008 and *p* = 0.015, respectively). No significant change was observed between T2 and T3.

The comparison of neurodegenerative biomarkers at T1, T2, and T3 between the groups is presented in Table 5, Figure 3. Compared to Group 2, the levels of PMAPT/pTau, S100B, NSE, and GFAP measured after preoxygenation, post-endotracheal intubation, and at the end of the surgical procedure were higher and statistically significant in Group 1 (*p* < 0.001 for all parameters). This suggests greater axonal and astrocyte damage in Group 1. The NfL level in Group 1 was lower than that in Group 2 at all time points, but this difference was not statistically significant.

## 4. Discussion

In our study, we investigated the acute effects of two different preoxygenation methods used in anesthesia on neurodegenerative biomarkers in patients undergoing laparoscopic cholecystectomy surgery. This is the first study to examine the acute impact of short-term oxygen therapy on neurodegenerative markers. We found that oxygen is more toxic to neurodegenerative biomarkers when administered via the traditional preoxygenation method, which involves a longer duration, compared to the rapid preoxygenation method.

It is well known that oxygen can be neurotoxic when administered at high concentrations. Previous research has shown that high concentrations of oxygen can increase the production of free radicals, leading to oxidative stress and potentially playing a role in the pathogenesis of neurodegenerative diseases [20]. Martin et al. have noted that oxygen, when delivered at a high FiO_2_ amount, can lead to oxidative stress and cellular damage [1]. Alva et al., who studied the cellular mechanisms of normobaric hyperoxia, found that high oxygen concentrations can have severe toxic effects on cells [3].

Research on the risks and benefits of preoxygenation suggests that high FiO_2_ concentrations may induce oxidative stress [8]. Ottolenghi et al. found that the use of a high FiO_2_ amount in anesthesia and intensive care can lead to oxidative stress and cellular damage by oxidizing ROS, lipids, DNA, and proteins [5]. Similarly, Nimmagadda et al. examined the physiological effects and potential risks of preoxygenation and argued that high oxygen levels could cause cellular damage [6]. The results of our study also suggest that high amounts of oxygen can cause toxicity even in the acute period. However, despite all these potential harms, preoxygenation should be performed in cases with anticipated difficult airways and in critically ill patients. Moreover, preoxygenation is recommended for all patients scheduled for intubation in anesthesia and intensive care [22,24].

In our study, we examined the neurodegenerative biomarkers pTau, S100B, NSE, NfL, and GFAP. All of these markers are considered biomarkers of central nervous system injury [10,11]. We focused on biomarkers indicating astrocyte damage (S100B, GFAP), neuronal damage (NSE), axonal damage (NfL, pTau), and glial cell damage (GFAP). To investigate the acute effects of oxygen toxicity on these biomarkers, we compared two different preoxygenation methods.

When comparing the groups, a significant increase in pTau levels was observed in Group 1. Our findings align with those of the study by Amoo et al., suggesting that high oxygen concentrations may lead to the hyperphosphorylation of tau proteins and neurodegeneration [10]. Similarly, the increase in S100B levels, which was more pronounced in Group 1, may indicate astrocyte damage, which is consistent with the findings of Hussein et al. [19].

Changes in NSE levels were evaluated as an indicator of neuronal damage. A significant increase in NSE levels was observed in Group 1 compared to Group 2, and this increase was statistically significant and consistent with the literature [10]. In line with our study, Peng et al. reported that exposure to a high FiO_2_ amount increases oxidative stress, inflammation, and apoptosis, leading to elevated S100B and NSE levels and potentially resulting in neurological damage [18].

NfL, a dynamic biomarker of axonal damage, is known to increase in the chronic phase [12,33]. In our study, no significant increase in NfL levels was observed in either group. We believe this is due to the fact that NfL is a marker that rises in the chronic phase, while our study focused on the acute phase [11].

GFAP is a biomarker indicating glial cell damage and may increase in response to oxidative stress. It is considered an important biomarker in neurodegenerative diseases such as Alzheimer’s disease [20,34]. Oxygen toxicity has been suggested to increase GFAP levels through astrogliosis and other inflammatory processes. Astrocytes may increase GFAP production to maintain homeostatic balance under oxidative stress conditions. This process leads to elevated GFAP levels as an indicator of neuroinflammation and neural cell damage [20,31]. In our findings, a significant increase in GFAP levels was observed in Group 1. These results are supported by the study conducted by Ganne et al. [20]. Additionally, the increase in GFAP levels in our study supports the notion that short-term oxygen exposure induces stress on glial cells. It suggests that short-term and high-dose oxygen exposure may be a risk factor for the development of neurodegenerative diseases such as Alzheimer’s in later life. Interestingly, in our study, GFAP levels decreased as FiO_2_ levels declined over time. This may indicate that astroglial damage decreased as the concentration of FiO_2_ used during the operation decreased over time.

The hemodynamic responses in patients may be altered due to surgical stress and tracheal intubation. The changes observed in our study probably reflect the physiological stress associated with surgical and anesthetic interventions [35]. The elevated HR and SAP levels at T1 and T2 indicate sympathetic activation and catecholamine release triggered by surgical stress and intubation. However, significant increases in HR, SAP, DAP, and MAP levels in Group I also suggest that standard preoxygenation with 100% O_2_ for 3 min may elicit a greater stress response. In contrast, Group II showed a more stable profile, with significant changes observed only in HR and SAP. These findings also suggest that rapid preoxygenation may reduce hemodynamic fluctuations by limiting oxygen exposure and oxidative stress. Optimizing preoxygenation strategies is particularly important for patients at risk of hemodynamic instability.

This study has enhanced our understanding of the effects of preoxygenation methods used in laparoscopic cholecystectomy on neurological biomarkers, providing valuable insights into the neurodegenerative risks associated with oxygen therapy. While no significant differences in duration were observed between the standard and rapid preoxygenation methods, both using 100% FiO_2_, our findings suggest that preoxygenation may exacerbate cellular damage, as reflected by elevated levels of neurodegenerative biomarkers. These results underscore the potential risks of preoxygenation and oxygen toxicity, highlighting the importance of caution in its clinical application.

Future research should focus on longitudinal studies to evaluate the long-term neurological effects of preoxygenation, particularly in high-risk populations such as geriatric and pediatric patients [7,8]. Long-term monitoring of biomarker dynamics and cognitive outcomes will provide deeper insights into these processes. Additionally, multimodal approaches, including advanced imaging techniques (e.g., brain MRI) and neurophysiological evaluations (e.g., EEG) could offer a more comprehensive understanding of the structural and functional impacts of oxygen therapy. Optimizing the duration, method, and FiO_2_ ratio of preoxygenation remains a critical area of research to balance therapeutic benefits with potential risks. Preoxygenation strategies must consider both the immediate and long-term neurological health of patients, and the integration of longitudinal and multimodal research designs will be essential to improving these strategies and advancing neuroprotective practices in anesthesia and surgery.

Our study has several limitations. First, the small sample size restricts the generalizability of our findings, particularly to older or high-risk populations. Secondly, the absence of preoperative baseline biomarker measurements before the stressful surgical interventions makes it challenging to distinguish the effects of preoxygenation methods from pre-existing conditions, necessitating a cautious interpretation of the study’s findings. Future studies should include baseline assessments to improve interpretability. Additionally, biomarkers were only assessed during the acute surgical phase, limiting insights into long-term effects. Long-term follow-up, particularly in geriatric patients, is necessary to evaluate potential chronic neurological impacts. The absence of multimodal assessments, such as MRI and EEG, further restricts the depth of our findings. Lastly, focusing on a limited set of biomarkers excludes other potentially relevant indicators of neuronal damage and stress.

Addressing these limitations in future studies, through larger sample sizes, baseline measurements, multimodal assessments, and long-term follow-up, will enhance our understanding and improve the clinical applicability of preoxygenation strategies in anesthesia and surgery, while our study provides foundational insights into their acute effects.

## 5. Conclusions

Our study is the first to compare the acute effects of two different preoxygenation methods on neurodegenerative biomarkers in laparoscopic cholecystectomy surgery. Our results indicate that preoxygenation with 100% FiO_2_ induces stress in neuronal cells, axons, and glial cells, leading to an increase in neurodegenerative biomarkers. Considering our findings, we recommend that preoxygenation should be used sparingly and with caution, particularly in patients with anticipated difficult airways, to minimize its potential harms. It is of great importance to conduct large-scale, randomized clinical trials to optimize the duration, method, and FiO_2_ concentration used in preoxygenation.

## Figures and Tables

**Figure 1 medicina-61-00167-f001:**
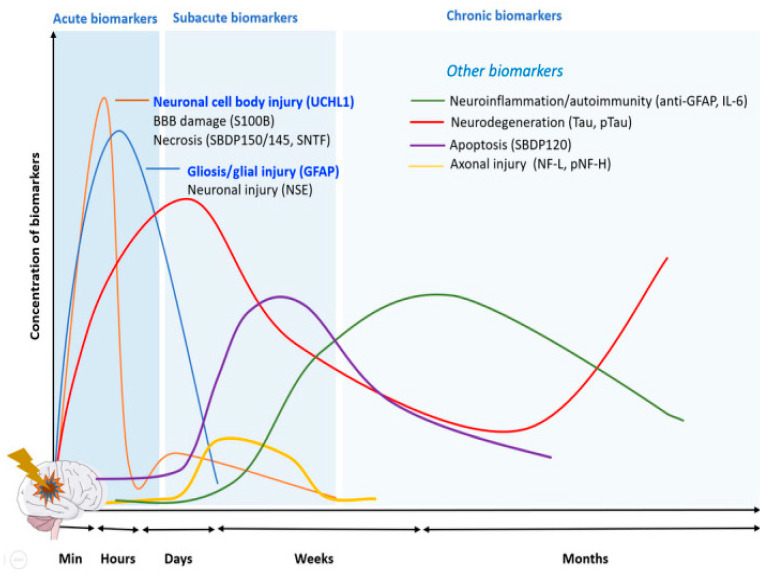
Temporal variation of neurodegenerative biomarkers [11].

**Figure 2 medicina-61-00167-f002:**
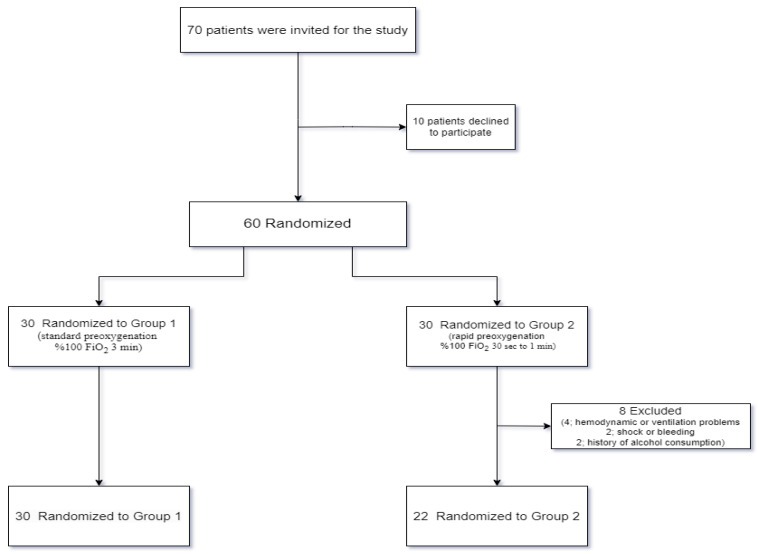
Flowchart of patient enrollment and admission (Group I: standard preoxygenation; Group II: rapid preoxygenation).

**Figure 3 medicina-61-00167-f003:**
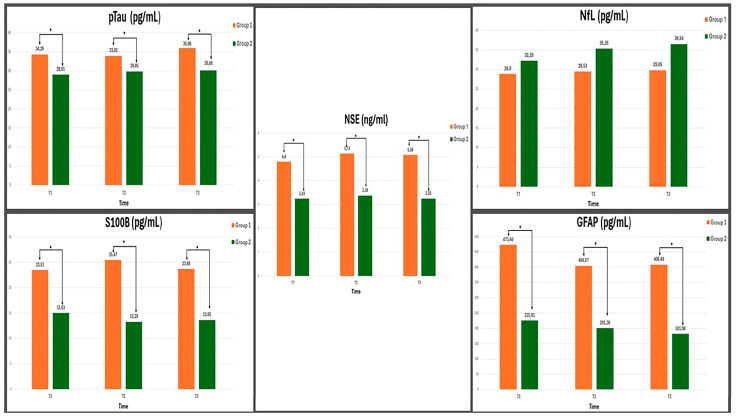
Temporal changes in neurodegeneration Biomarkers in Group 1 and Group 2 (* Group 1 vs. Group 2).

**Table 1 medicina-61-00167-t001:** Demographics and baseline patient characteristics between the two groups.

	Group 1 (30)	Group 2 (22)	*p*
Age	44.83 ± 13.79	39.22 ± 13.34	0.093
Gender (F/M)	26/4	9/3	0.641
Smoking (+)	7	11	0.176
Medication (+)	5	1	0.243
Anesthesia Duration	55.80 (20)	65.40 (45.5)	0.321
Surgery Duration	42.03 (11.5)	55.00 (33.75)	0.097
Chronic Disease (52)			
None	23	18	
Hypertension (HT)	3	3	
Chronic Obstructive Pulmonary Disease (COPD)	2	0	
Hyperthyroidism	1	0	
HT + DM + KOAH	1	0	
HT + KOAH	0	1	

**Table 2 medicina-61-00167-t002:** Baseline laboratory data between the two groups.

	Group 1 (30)	Group 2 (22)	*p*
Glucose (mg/dL)	100.86 ± 19.99	96.13 ± 12.36	0.415
Na (mmol/L)	141.12 ± 2.60	140.15 ± 2.15	0.154
K (mmol/L)	4.12 ± 0.76	4.14 ± 0.54	0.904
Urea (mg/dL)	27.28 ± 8.71	24.59 ± 6.99	0.243
Creatinine (mg/dL)	0.67 ± 0.14	0.69 ± 0.10	0.593
AST (U/L)	27.06 ± 14.65	24.09 ± 15.99	0.308
ALT (U/L)	28.36 ± 24.71	33.72 ± 44.01	0.97
GGT (U/L)	45.20 ± 55.10	52.50 ± 56.35	0.753

**Table 3 medicina-61-00167-t003:** Comparison of hemodynamic parameters between groups and over time.

Time		T1	T2	T3	*p*
HR	Group 1	86.26 ± 13.69 *	86.76 ± 19.60 ^#^	76.03 ± 15.37	0.001
	Group 2	87.45 ± 16.38	90.81 ± 10.74 ^#^	82.22 ± 14.06	0.029
SpO_2_	Group 1	98.80 (2)	99.03 (2)	99.16 (1)	0.541
	Group 2	98.54 (1.25)	98.77 (0.25)	98.86 (0)	0.494
SAP	Group 1	137.83 ± 22.97 ^&,^*	124.50 ± 24.24	123.56 ± 21.67	0.002
	Group 2	130.72 ± 24.14 ^&^	114.13 ± 9.99	116.22 ± 15.44	0.013
DAP	Group 1	82.60 (22.25) *	81 (27.75)	73.50 (27.25)	0.022
	Group 2	77.81 (19)	68 (18)	71 (16.50)	0.357
MAP	Group 1	103.66 (29.75) *	98.50 (32.50)	94.50 (31.50)	0.015
	Group 2	95.68 (15.5)	83.50 (14)	86.50 (13)	0.062

(* T1 vs. T3, ^#^ T2 vs. T3, ^&^ T1 vs. T2).

**Table 4 medicina-61-00167-t004:** Temporal changes in neurodegeneration biomarkers within themselves (* T1 vs. T2, T3; ^#^ T1 vs. T2, T3).

Time	Group 1 (30)	T1	T2	T3	*p*
	Group 2 (22)				
pTau (pg/mL)	Group 1	34.29 (2.31)	33.92 (1.56)	35.96 (2.52)	0.394
	Group 2	29.01 (1.73)	29.85 (3.18)	30.08 (2.679	0.292
S100B (pg/mL)	Group 1	23.51 (10.25)	25.47 (14.47)	23.68 (12.94)	0.645
	Group 2	15.03 (3.58)	13.29 (3.98)	13.64 (6.32)	0.784
NSE (ng/mL)	Group 1	4.92 (1.11)	5.13 (1.16)	5.08 (1.41)	0.141
	Group 2	3.22 (0.25)	3.38 (0.37)	3.25 (0.2)	0.178
NfL (pg/mL)	Group 1	28,87 (14,76)	29.52 (15.53)	28.94 (18.86)	0.827
	Group 2	32.28 (14.23)	35.35 (10.67)	34.24 (13.82)	0.331
GFAP (pg/mL)	Group 1	468.02 (277.48) *	404.87 (149.66)	382.88 (185.32)	0.011 *
	Group 2	225.91 (107.55) ^#^	201.36 (124.51)	160.62 (93.91)	0.011 *

**Table 5 medicina-61-00167-t005:** Temporal changes in neurodegeneration biomarkers in Group 1 and Group 2 (Group 1 vs. Group 2).

Time	pTau (pg/mL)			S100B (pg/mL)			NSE (ng/mL)			NfL (pg/mL)			GFAP (pg/mL)		
	**Group 1**	**Group 2**	** *p* **	**Group 1**	**Group 2**	** *p* **	**Group 1**	**Group 2**	** *p* **	**Group 1**	**Group 2**	** *p* **	**Group 1**	**Group 2**	** *p* **
T1	34.29 (2.31)	29.01 (1.73)	0.001	23.51 (10.25)	15.03 (3.58)	0.001	4.92 (1.11)	3.22 (0.25)	0.001	28.87 (14.76)	32.28 (14.23)	0.317	468.02 (277.48)	225.91 (107.55)	0.001
T2	33.92 (1.56)	29.85 (3.18)	0.001	25.47 (14.47)	13.29 (3.98)	0.001	5.23 (1.16)	3.28 (0.37)	0.001	29.52 (15.53)	35.35 (10.67)	0.113	404.87 (149.66)	201.36 (124.51)	0.001
T3	35.96 (2.52)	30.08 (2.67)	0.001	23.68 (12.94)	13.64 (6.32)	0.001	4.91 (1.41)	3.19 (0.20)	0.001	29.85 ± 10.67	36.54 ± 12.23	0.826	408.49 ± 123.07	181.98 ± 81.62	0.031

(pTau; Indicates axon damage, S100B; Indicates astrocyte damage, NSE; Indicates body damage, NfL; Indicates axon damage, GFAP; Indicates astrocyte damage).

## Data Availability

The datasets used and/or analyzed during the current study are available from the corresponding author on reasonable request.

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
