# Peer review of "Investigation of the Acute Effects of Two Different Preoxygenation Methods on Neurodegenerative Biomarkers in Laparoscopic Cholecystectomy Surgery"

_medicina, 2025, doi:10.3390/medicina61020167_

Round 1

Reviewer 1 Report

Comments and Suggestions for Authors

Thank you for the opportunity to review such an important and critical original study, on the toxic effects of hyperoxygenation during anesthesia. This is definitely a well-designed and carefully written study that is worthwhile to be published.

Some minor remarks for authors:

1)    Abstract: Please add a full stop after “over 30 seconds to 1 minute”

2)    Figure1: Please elaborate a little bit on the Figure and on the different expected actions of these biomarkers, so that the reader be better oriented.

3)    Introduction section: Please add a citation after the sentence “Anesthesia is often viewed as the primary culprit….the target organ”

4)    Limitations section: Please add on the limitations sections that all of these markers were not measured in baseline conditions for patients, prior to the stressful surgical interventions, so the results of the study should be considered with caution.

5)    Have you considered recording patients’ morbidity or mortality after surgery and hyperoxygenation, which could be central endpoints for making conclusions?

I am looking forward to your response, and I hope that this study could be further expanded.

Author Response

We sincerely thank the reviewers for their thoughtful and constructive feedback on our manuscript. Your detailed comments and suggestions have been invaluable in improving the clarity, scientific rigor, and overall quality of our work. We have carefully addressed each of your concerns and incorporated the necessary revisions into the manuscript. We believe these changes have strengthened the study and enhanced its contribution to the field. We greatly appreciate your time, expertise, and dedication in reviewing our manuscript.

Comments 1: Abstract: Please add a full stop after “over 30 seconds to 1 minute”

Response 1: We sincerely thank you for your meticulous review and for highlighting the need for clarity in the phrase "30 seconds to 1 minute." We have revised this by adding a period at the end of the phrase to enhance readability and precision. We greatly appreciate your attention to detail and believe that this adjustment significantly contributes to the overall clarity of our manuscript.

Comments 2: Figure1: Please elaborate a little bit on the Figure and on the different expected actions of these biomarkers, so that the reader be better oriented.

Response 2: We sincerely thank you for your thoughtful suggestion to provide additional details about Figure 1 and the expected actions of the biomarkers presented. In response, we have revised the figure legend to include more comprehensive information about the roles of these biomarkers and their anticipated changes at different stages. These adjustments aim to better guide readers and enhance their understanding of the data. We greatly appreciate your valuable feedback, which has helped us improve the clarity and scientific rigor of our manuscript.

Comments 3:    Introduction section: Please add a citation after the sentence “Anesthesia is often viewed as the primary culprit….the target organ”

Response 3: Thank you for your insightful comment regarding the need for a citation to support the statement about anesthesia’s role in cognitive decline. We have revised the sentence in the introduction and added a relevant citation to address this concern.

The revised sentence now reads:
“Although anesthetic agents are often regarded as the primary contributors to cognitive decline due to their direct effects on the brain, oxygen has also emerged as a significant factor in this process [25].”

The added reference is as follows:
[25] Zhang M, Yin Y. Dual roles of anesthetics in postoperative cognitive dysfunction: Regulation of microglial activation through inflammatory signaling pathways. Front Immunol 2023;14. https://doi.org/10.3389/fimmu.2023.1102312

We appreciate your attention to detail and believe this revision strengthens the scientific foundation of our manuscript.

Comments 4:    Limitations section: Please add on the limitations sections that all of these markers were not measured in baseline conditions for patients, prior to the stressful surgical interventions, so the results of the study should be considered with caution.

Response 4: Thank you for highlighting the importance of including a discussion about the absence of preoperative baseline biomarker measurements in the limitations section. We have addressed this by adding the following statement to the limitations:

“Secondly, the absence of preoperative baseline biomarker measurements before the stressful surgical interventions makes it challenging to distinguish the effects of preoxygenation methods from pre-existing conditions, necessitating a cautious interpretation of the study's findings.”

This revision has been incorporated into the manuscript to ensure transparency and to acknowledge this important limitation. We sincerely appreciate your thoughtful feedback, which has helped us improve the clarity and comprehensiveness of the limitations section.

Comments 5:    Have you considered recording patients’ morbidity or mortality after surgery and hyperoxygenation, which could be central endpoints for making conclusions?

Response 5: Thank you for your insightful suggestion to include postoperative morbidity, mortality, and hyperoxygenation as potential endpoints. While these are indeed critical outcomes in many clinical studies, the primary aim of our study was to investigate the acute effects of preoxygenation methods on neurodegenerative biomarkers during laparoscopic cholecystectomy. As such, our study was not designed to assess long-term clinical outcomes like morbidity or mortality.

However, we agree that incorporating such endpoints in future studies would provide a more comprehensive understanding of the clinical implications of preoxygenation strategies. Monitoring postoperative morbidity and mortality, along with hyperoxygenation-related effects, could offer valuable insights into the broader impact of oxygen therapy and its optimization. We appreciate your valuable feedback, which has inspired us to consider these important factors in the design of future research.

Reviewer 2 Report

Comments and Suggestions for Authors

The study presents a notable innovation as, to the best of our knowledge, it is the first to investigate the acute effects of different preoxygenation methods on neurodegenerative biomarkers during laparoscopic cholecystectomy surgery. This approach highlights the potential neurotoxic effects of 100% FiO₂ preoxygenation, a routine yet often overlooked anesthesia practice. Strengths include its prospective, randomized controlled design, the use of validated biomarkers, and clear clinical relevance in optimizing preoxygenation strategies to minimize neurological risks.

However I have some serious concerns in relation to the manuscript.

Major Concerns

  1. Sample Size and Generalizability: The small sample size (30 in Group I, 22 in Group II) and unequal group distribution may affect statistical power and reliability. Additionally, the population—relatively young with minimal comorbidities—limits generalizability to broader clinical contexts, such as older or high-risk patients. The absence of data on participants’ ethnicity also restricts the applicability of findings, as biomarker baselines may vary demographically.

  2. Reference Values and Relevance: The lack of reference values for biomarkers (pTau, S100B, NSE, GFAP, and NfL) in healthy young populations complicates the interpretation of results, as most studies focus on older cohorts. It is unclear if the observed changes signify meaningful neurological stress or natural variability. The magnitude of biomarker fluctuations may lack clinical relevance, especially given the acute nature of the study and the age range of participants, where transient changes are unlikely to have long-term consequences.

  3. Multimodal and Longitudinal Assessments: A multimodal approach incorporating quantitative brain MRI (e.g., volumetry, cortical thickness), quantitative EEG, and neurophysiological evaluations would provide deeper insights into neurological impacts. Additionally, longitudinal assessments of pre- and post-operative neurological and neuropsychological status would clarify whether biomarker changes correlate with clinical outcomes. Long-term biomarker monitoring is also essential to determine if observed differences persist or resolve, as isolated changes in young, low-risk populations are unlikely to hold lasting clinical significance.

Minor Suggestions

The introduction could be improved by providing a clearer clinical context, emphasizing the relevance of biomarkers in anesthetic practice and neuroprotection. A broader review of the literature on oxygen-related oxidative stress and neurological damage would provide additional depth. Explicitly stating the hypothesis and strengthening the justification for the biomarkers selected, particularly in a young surgical population, would improve focus.

The discussion would benefit from a more thorough clinical interpretation of the results, linking biomarker changes to oxidative stress and their practical implications. Comparisons with previous studies and an exploration of potential mechanisms should be included. Limitations, such as the lack of long-term follow-up, absence of multimodal assessments (e.g., MRI, EEG), and limited generalizability, should be critically addressed. Proposing longitudinal and multimodal studies as future research directions would enhance the discussion’s impact.

Replacing extensive tables with visual graphs could also improve clarity and accessibility. Boxplots can show biomarker distributions across groups and time points, while line graphs can illustrate temporal trends (T1, T2, T3), improving data visualization and reader comprehension.

Author Response

We sincerely thank you for your comprehensive evaluation and insightful comments on our manuscript. Your constructive feedback has been invaluable in improving our study and enhancing the clarity, scientific rigor, and overall quality of the manuscript. We have carefully addressed each suggestion and incorporated them into the revised version of the manuscript. Below, we provide detailed responses to each of your comments and outline the corresponding changes made to the manuscript. We deeply appreciate the time and effort you have devoted to providing such thoughtful and detailed feedback. Thank you once again for your valuable input.

Major Concerns

Comments 1: Sample Size and Generalizability: The small sample size (30 in Group I, 22 in Group II) and unequal group distribution may affect statistical power and reliability. Additionally, the population—relatively young with minimal comorbidities—limits generalizability to broader clinical contexts, such as older or high-risk patients. The absence of data on participants’ ethnicity also restricts the applicability of findings, as biomarker baselines may vary demographically.

Response 1: Dear Reviewer,

We appreciate your thoughtful critique regarding the sample size and characteristics of the study population, and we thank you for raising these important points. Below, we address this critique under three subheadings:

(A) Sample Size:
We acknowledge that the relatively small sample size (Group I: 30, Group II: 22) and unequal group distribution may affect statistical power and reliability. However, our study serves as the first investigation into the acute effects of preoxygenation methods on neurodegenerative biomarkers. To address your concerns, we present the results of a post-hoc G-power analysis based on the pTau value at T1. This analysis, conducted with a significance level of 0.05 and a total sample size of 52 (n1=30, n2=22), calculated an effect size of 2.58, resulting in a power of 1.00. These results indicate that our study was sufficiently powered to detect significant differences in primary outcomes, mitigating concerns about statistical reliability.

That said, we recognize that larger and more evenly distributed groups would further enhance the robustness and applicability of our findings to the general patient population. The trends observed in this controlled setting provide valuable preliminary data to guide future research with larger, more balanced cohorts to confirm and expand on our findings. Despite the small sample size, we employed robust statistical methods to minimize bias and ensure the reliability of our results.

(B) Young Patient Population:
Our focus on a young and low-risk patient population was intentional, as it allowed us to reduce confounding variables such as age-related comorbidities and baseline variations in biomarkers. This approach enabled a clearer evaluation of the physiological effects of different preoxygenation methods. However, we acknowledge that these findings may not be directly generalizable to older or high-risk patients, who may respond differently to such interventions. Furthermore, we chose to study a young patient population because potential agents like oxygen, which could cause neurodegenerative damage, might pose risks for comorbidities in their future lives. Future research involving more diverse populations with broader clinical risk profiles is essential to evaluate the applicability of our findings to more complex patient groups.

(C) Ethnicity Data:
We agree with the reviewer that biomarker baselines can vary demographically, particularly with respect to ethnicity. While ethnicity data were not collected in this study, we recognize its importance and will incorporate this aspect into future research to enhance the generalizability and clinical applicability of our findings.

Although our study has limitations, it provides a foundational understanding of the acute effects of preoxygenation methods. Your feedback is invaluable in guiding future studies and refining our research design to address these critical questions.

Comments 2: Reference Values and Relevance: The lack of reference values for biomarkers (pTau, S100B, NSE, GFAP, and NfL) in healthy young populations complicates the interpretation of results, as most studies focus on older cohorts. It is unclear if the observed changes signify meaningful neurological stress or natural variability. The magnitude of biomarker fluctuations may lack clinical relevance, especially given the acute nature of the study and the age range of participants, where transient changes are unlikely to have long-term consequences.

Response 2: We thank the reviewer for their thoughtful critique regarding the lack of reference values for biomarkers in healthy young populations and its impact on result interpretation.

We acknowledge that the absence of established reference ranges for biomarkers such as pTau, S100B, NSE, GFAP, and NfL is a limitation. However, it is important to note that these parameters are still in the research phase, and no clinical reference ranges have been established for any age group. Most existing studies focus on older or high-risk populations, making direct comparisons challenging. For this reason, our study was designed to include control groups, with the primary aim of comparing the advantages and effects of two different preoxygenation methods.

By focusing on a younger population, we aimed to observe the effects of these biomarkers without the confounding influence of chronic conditions often present in older individuals, which could affect brain function. Our study provides valuable preliminary data by documenting biomarker fluctuations in a younger cohort under controlled conditions.

We agree, however, that future research should aim to establish baseline reference ranges for these biomarkers across various age groups to enhance the interpretability and clinical applicability of such findings.

Comments 3: Multimodal and Longitudinal Assessments: A multimodal approach incorporating quantitative brain MRI (e.g., volumetry, cortical thickness), quantitative EEG, and neurophysiological evaluations would provide deeper insights into neurological impacts. Additionally, longitudinal assessments of pre- and post-operative neurological and neuropsychological status would clarify whether biomarker changes correlate with clinical outcomes. Long-term biomarker monitoring is also essential to determine if observed differences persist or resolve, as isolated changes in young, low-risk populations are unlikely to hold lasting clinical significance.

Response 3:Dear Reviewer,

We sincerely thank you for your insightful comments and valuable suggestions regarding the inclusion of multimodal approaches and longitudinal assessments to enhance the depth and clinical relevance of our study. Below, we address your critique under three subheadings:

(A) Multimodal Assessments:
The study was designed to predict metabolic damage, and parameters reflecting macro-level damage, such as brain MRI and EEG, were not included. While we acknowledge that incorporating these parameters could have added value to our findings, budgetary constraints prevented their inclusion. This limitation is recognized and duly noted. We agree that including these suggested parameters could further enhance the robustness and clinical impact of future studies.

(B) Longitudinal Neurological and Neuropsychological Evaluations:
We recognize the importance of longitudinal evaluations in determining whether biomarker changes translate into meaningful clinical outcomes. While tools and scales used for neurological and neuropsychological assessments are effective for identifying long-term neurological damage, they may be insufficient for predicting acute effects in the short term. Given that our study was designed to focus on the acute effects of oxygen, the inclusion of such evaluations was not deemed necessary.

(C) Long-Term Biomarker Monitoring:
We agree with the reviewer that the transient nature of biomarker changes in young, low-risk populations may limit their immediate clinical significance. Long-term monitoring is indeed essential to determine whether the observed differences persist or resolve over time. However, our study’s hypothesis did not include a long-term follow-up process, as it was specifically designed to examine acute effects. The nature of laparoscopic cholecystectomy procedures involves short hospital stays, and due to the socio-cultural context of the region where the study was conducted, follow-up with patients postoperatively posed significant challenges. These factors represent limitations of our study.

We greatly appreciate your thoughtful feedback, which provides a foundation for improving the design and scope of future research.

Minor Suggestions

Dear Reviewer, 

We sincerely thank you for your constructive and thoughtful comments, which provide valuable guidance for improving our manuscript. Below, we address each of the suggested areas under the provided headings:

Comments 1: The introduction could be improved by providing a clearer clinical context, emphasizing the relevance of biomarkers in anesthetic practice and neuroprotection. A broader review of the literature on oxygen-related oxidative stress and neurological damage would provide additional depth. Explicitly stating the hypothesis and strengthening the justification for the biomarkers selected, particularly in a young surgical population, would improve focus.

Response 1: We agree that the introduction would benefit from a clearer clinical context emphasizing the importance of biomarkers in anesthetic practice and neuroprotection. In response to your recommendations, the revised manuscript includes these necessary adjustments, and the introduction has been updated and reuploaded. These changes provide a more focused and comprehensive foundation for the study.

Comments 1:The discussion would benefit from a more thorough clinical interpretation of the results, linking biomarker changes to oxidative stress and their practical implications. Comparisons with previous studies and an exploration of potential mechanisms should be included. Limitations, such as the lack of long-term follow-up, absence of multimodal assessments (e.g., MRI, EEG), and limited generalizability, should be critically addressed. Proposing longitudinal and multimodal studies as future research directions would enhance the discussion’s impact.

Response 2: We appreciate your suggestion to deepen the clinical interpretation of our findings. Comparisons with previous studies, the limitations of long-term follow-up and multimodal assessments, and the clinical significance of these aspects for future research have been addressed. In line with your recommendations, the discussion section has been revised and reuploaded with these adjustments.

Comments 3:Replacing extensive tables with visual graphs could also improve clarity and accessibility. Boxplots can show biomarker distributions across groups and time points, while line graphs can illustrate temporal trends (T1, T2, T3), improving data visualization and reader comprehension.

Response 3: We acknowledge that visual graphics can enhance data clarity and accessibility. In the revised manuscript, we have replaced some detailed tables with box plots to illustrate biomarker distributions across groups and time points (T1, T2, T3). Additional graphs have been incorporated alongside tables as per your suggestions, and the updated version has been reuploaded.

We greatly appreciate your feedback, which has significantly improved the quality of our manuscript and its presentation.

Conclusion:

Dear Reviewer,

We sincerely thank you for your thoughtful and constructive feedback, which has highlighted key areas for improvement in our study. Incorporating these valuable critiques and suggestions will significantly strengthen our manuscript and contribute to the ongoing discussions on preoxygenation strategies, neuroprotection, and the use of biomarkers in anesthetic practice.

We are truly grateful for the time and effort you have dedicated to providing this invaluable feedback. Your constructive comments will help ensure that our work meets the highest standards of scientific rigor and impact, ultimately enhancing the scientific value of our manuscript.

Kind regards...

Round 2

Reviewer 2 Report

Comments and Suggestions for Authors

Authors have been able to address most of my concerns and significantly have improved the manuscript explaining main limitations and research gaps to be solved in the future. My main concern continues to be the small sample size and lack of cut-off points of the analytical parameters used, both aspects reduce the potential aplication of the obtained results, but for a discovery first work could be acceptable.